# A machine learning approach to brain epigenetic analysis reveals kinases associated with Alzheimer's disease

Yanting Huang[1,11], Xiaobo Sun[2,11,12✉], Huige Jiang[3], Shaojun Yu[1], Chloe Robins[4], Matthew J. Armstrong[5], Ronghua Li[5], Zhen Mei[4], Xiaochuan Shi[6], Ekaterina Sergeevna Gerasimov [4], Philip L. De Jager [7], David A. Bennett[8], Aliza P. Wingo [9,10], Peng Jin [5], Thomas S. Wingo [4,5,12✉] & Zhaohui S. Qin [3,12✉]

Alzheimer's disease (AD) is influenced by both genetic and environmental factors; thus, brain epigenomic alterations may provide insights into AD pathogenesis. Multiple array-based Epigenome-Wide Association Studies (EWASs) have identified robust brain methylation changes in AD; however, array-based assays only test about 2% of all CpG sites in the genome. Here, we develop EWASplus, a computational method that uses a supervised machine learning strategy to extend EWAS coverage to the entire genome. Application to six AD-related traits predicts hundreds of new significant brain CpGs associated with AD, some of which are further validated experimentally. EWASplus also performs well on data collected from independent cohorts and different brain regions. Genes found near top EWASplus loci are enriched for kinases and for genes with evidence for physical interactions with known AD genes. In this work, we show that EWASplus implicates additional epigenetic loci for AD that are not found using array-based AD EWASs.

[1] Department of Computer Science, Emory University, Atlanta, GA, USA. [2] Department of Mathematical and Statistical Finance, School of Statistics and Mathematics, Zhongnan University of Economics and Laws, Wuhan, Hubei, China. [3] Department of Biostatistics and Bioinformatics, Rollins School of Public Health, Emory University, Atlanta, GA, USA. [4] Department of Neurology, Emory University School of Medicine, Atlanta, GA, USA. [5] Department of Human Genetics, Emory University School of Medicine, Atlanta, GA, USA. [6] Department of Statistics, University of Washington, Seattle, WA, USA. [7] Center for Translational and Computational Neuroimmunology, Department of Neurology, Columbia University Medical Center, New York, NY, USA. [8] Rush Alzheimer's Disease Center, Rush University Medical Center, Chicago, IL, USA. [9] Division of Mental Health, Atlanta VA Medical Center, Decatur, GA, USA. [10] Department of Psychiatry, Emory University School of Medicine, Atlanta, GA, USA. [11] These authors contributed equally: Yanting Huang, Xiaobo Sun. [12] These authors jointly supervised this work: Xiaobo Sun, Thomas S. Wingo, Zhaohui S. Qin. ✉email: xsun28@gmail.com; thomas.wingo@emory.edu; zhaohui.qin@emory.edu

Alzheimer's disease (AD) is an age-dependent, neurodegenerative disorder, the leading cause of dementia, and a major public health concern world-wide[1]. AD is a complex illness due to environmental and genetic factors with a heritability of ~70%[2,3]. Compared to genome-wide association studies (GWASs), there are relatively fewer studies examining AD-associated epigenetic changes in the human brain. Yet, understanding epigenetic changes in the brain is important because they will likely illuminate both heritable and environmental aspects of AD pathogenesis. One of the most well-described epigenetic changes, DNA methylation (DNAm), is strongly linked with transcription regulation[4], is heritable[5], and notably changes in response to environmental exposure[6–8], such as smoking[9,10]. Important for AD and other age-dependent illnesses, it is also known to change with age[11].

Epigenome-wide association studies (EWASs) use array-based assays to test whether DNAm at particular CpG sites (abbreviated CpGs hereafter) is associated with a disease[12–15]. Multiple AD EWASs have identified differential DNAm associated with AD in different regions of the human brain, including prefrontal cortex (PFC)[16], entorhinal cortex (EC), superior temporal gyrus (STG), cerebellum (CER)[17], temporal pole region, temporal cortex, glia, neuron nuclei, non-neuronal nuclei[18], and superior temporal gyrus[19]. These works revealed AD-associated differential DNAm such as those near ANK1[16,19] and CDH23[16,17], which are distinct from AD GWAS signals. Although these studies have identified new AD-associated genes, array-based methods are limited because they only test about 2–3% of all CpGs in the human genome and have known technical limitations[20]. To overcome these challenges, we tested whether a machine learning approach could be used to identify additional AD-associated CpGs on a genome-wide scale.

In this work, we construct a supervised machine learning (ML) binary classifier named EWASplus to identify CpGs associated with AD. Given that epigenetic features and DNAm status are interconnected, we hypothesize that we can identify AD-associated CpGs using genomic and epigenetic features. Training data are derived from array-based EWASs, and features include relevant genomic and epigenomic profiling data (e.g., chromatin accessibility, histone modifications). After model training, we apply the trained model to the entire genome to identify additional AD-associated CpGs. Finally, we perform targeted bisulfite sequencing experiments to validate our in silico predictions. We find the highest rate of AD association for regions harboring putative CpGs predicted by EWASplus (65.8%; 25 out of 38), follow by CpGs known to associate with AD by methylation arrays (60.0%; 6 out of 10). Experimental validation shows predicted CpGs are 2.2 times more likely to be associated with AD ($p < 1.00 \times 10^{-9}$) than negative control CpGs. These results suggest EWASplus is capable of providing credible information to identify additional AD-associated CpGs.

## Results

**EWASplus overview.** The goal of EWASplus is to identify additional disease-associated CpGs that are not included on the methylation arrays. Currently, the most popular methylation arrays only represent 2–3% of all CpGs in the human genome. EWASplus aims to increase the number of CpGs tested in EWASs to a genome-wide scale. A comparison of CpG coverage between the 450K methylation array and EWASplus is shown in Supplementary Fig. 1. Standard EWAS operates under a testing framework, but EWASplus frames the problem as a supervised learning (i.e., classification) framework. The EWASplus approach (Fig. 1) is to (1) use summary statistics from array-based EWASs to classify all CpGs on the array into either trait-associated (positive)

or neutral (negative) group; (2) perform feature selection to identify the most informative features from a collection of 2256 genomic and epigenomic annotations; (3) train an ensemble learning model capable of identifying CpGs for trait association; and (4) score all CpGs in the entire genome to identify additional trait-associated CpGs not present on the array.

To prepare the training set, EWASplus gathers the most significant CpGs identified from array-based EWAS to form a positive training set. To reflect the fact that there are far fewer significant trait-associated CpGs in the genome than the trait-neutral ones, EWASplus selects a matching negative training set with similar genomic context that is ten times larger than the positive training set.

EWASplus employs an ensemble learning strategy and four different methods were chosen as the base learner: regularized logistic regression (RLR), support vector machine (SVM) classifier, random forest (RF), and gradient boosting decision trees (GBDT). To identify the best ensemble model, we tested all possible combinations of these base learners and found that the combination of RLR and GBDT gives the best performance overall (Supplementary Tables 1–6), and hence was selected to be the ensemble model in this study. RLR has the best recall but relatively low precision, while GBDT has the best precision but relatively low recall. When these two models are ensembled, the underfitting property of RLR can effectively offset the overfitting from GBDT while still keeping enough model complexity. More detailed description can be found in "Performance evaluation metrics" in the Methods section.

EWASplus can be applied to any array-based EWAS to extend its coverage. In this study, we tested EWASplus on data collected from four different cohorts: ROS/MAP (sample size 717), London (sample size 113), Mount Sinai (sample size 146), and Arizona (sample size 302). Cohort characteristics are given in Supplementary Table 7. All original EWASs were performed using the Illumina 450K methylation array.

**EWASplus performance compared to methylation array.** To evaluate the performance of EWASplus, we first considered its performance on CpGs present on the Illumina 450K methylation array (henceforth referred to as the "array"). Given the large sample size ($n = 717$), we choose data from the ROS/MAP cohort as the main dataset for performance evaluation. Methylation is measured on DNA derived from post-mortem PFC. Standard EWAS were conducted on six different AD-related traits: beta-amyloid density, Braak staging, the Consortium to Establish a Registry for Alzheimer's Disease (CERAD) score, cognitive trajectory, global AD pathology, and neurofibrillary tangle density (Supplementary Table 8). We trained a separate classifier for each of the six traits.

EWASplus results are summarized in Table 1 (see section "Hyperparameter tuning and ensemble model" for detailed description of the approach of evaluation). The area under the receiver operator characteristic (ROC) curve (AUC) values from the six traits range from 0.831 (cognitive trajectory) to 0.962 (neurofibrillary tangles) (Fig. 2a). The area under the precision-recall curve (PRC) (AUPRC) values from the six traits range from 0.502 (CERAD) to 0.858 (neurofibrillary tangles) (Fig. 2b). These results indicate that EWASplus works well to predict significant AD-associated CpGs for methylation measured by the array. Among the six traits, we observe the best performance for neurofibrillary tangles.

To further evaluate EWASplus, we asked whether the EWASplus prediction score is capable of distinguishing CpGs with differential DNAm between AD case and control status. To answer this question, we selected four groups of CpGs that differ

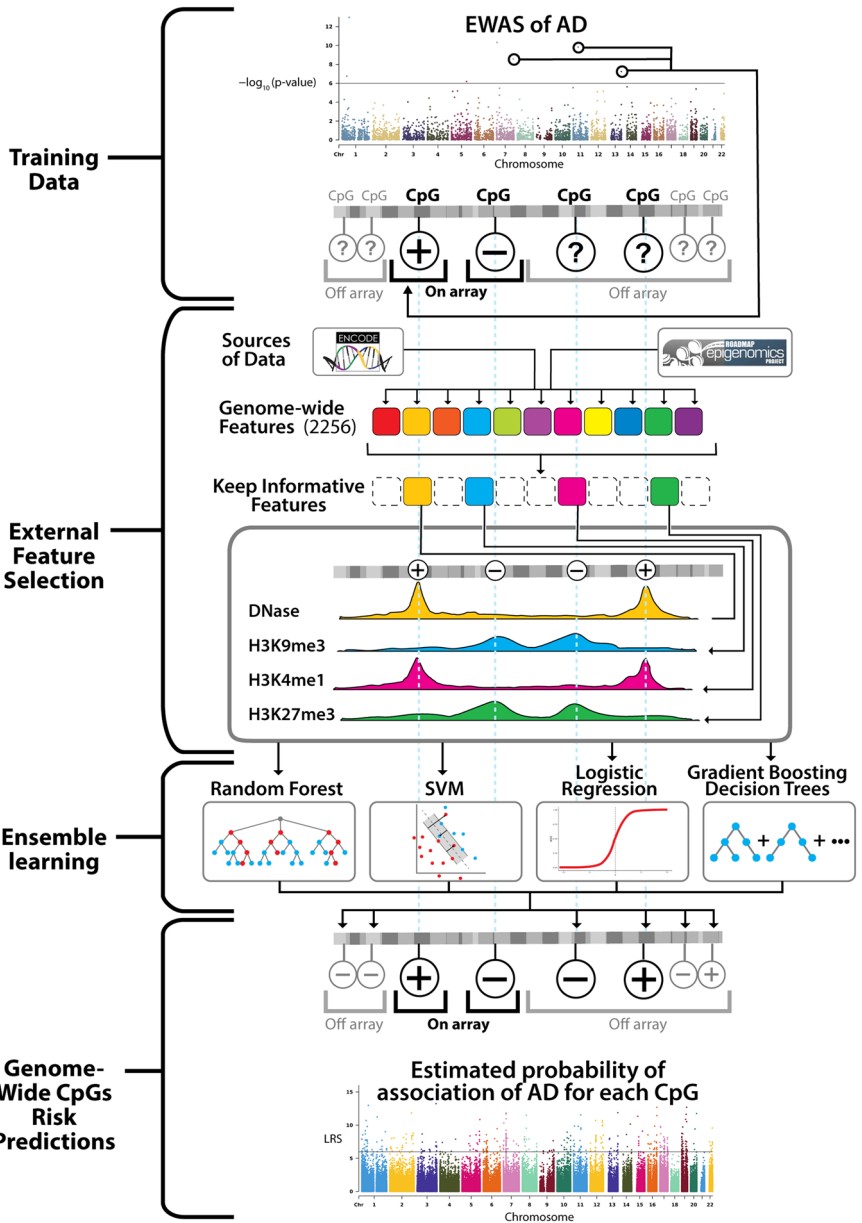

**Fig. 1 Overview of EWASplus approach.** The EWASplus procedure is composed of four major steps: (1) Training data collection from existing EWASs; (2) External feature (from sources such as ENCODE and Roadmap Epigenome consortia) selection; (3) Ensemble learning; and (4) Genome-wide CpGs risk prediction, in which trained ensemble learning model is applied genome-wide to score all CpGs.

**Table 1 Summary of performance evaluation of all six AD-related traits.**

| Outcome | Outcome type | AUC | AUPR | F1 | Precision | Recall |
|---|---|---|---|---|---|---|
| Beta-amyloid | Pathologic, IHC | 0.850 | 0.539 | 0.492 | 0.423 | 0.589 |
| Braak staging | Pathologic, Silver Stain | 0.860 | 0.599 | 0.530 | 0.487 | 0.581 |
| CERAD | Pathologic, Silver Stain | 0.833 | 0.502 | 0.508 | 0.457 | 0.571 |
| Cognitive trajectory | Clinical | 0.831 | 0.591 | 0.516 | 0.451 | 0.604 |
| Global pathology | Pathologic, Silver Stain | 0.882 | 0.622 | 0.577 | 0.507 | 0.671 |
| Neurofibrillary tangles | Pathologic, IHC | 0.962 | 0.858 | 0.754 | 0.677 | 0.852 |

The performance evaluation is on independent testing set on 450K array. The result reported here is in imbalanced setting (positive to negative CpGs ratio 1:10), which is closer to the real imbalanced setting in the human genome.

with respect to differential DNAm association with AD: (a) AD-associated CpGs in the positive training set (i.e., p-value less than the EWAS threshold); (b) CpGs suggestively associated with AD (i.e., p-value slightly greater than the EWAS threshold); (c) CpGs not associated with AD but not in negative training set; and (d) CpGs not associated with AD and in the negative CpG training set (i.e., p-value greater than the EWAS threshold and in negative training set). On average, we find a significant difference for

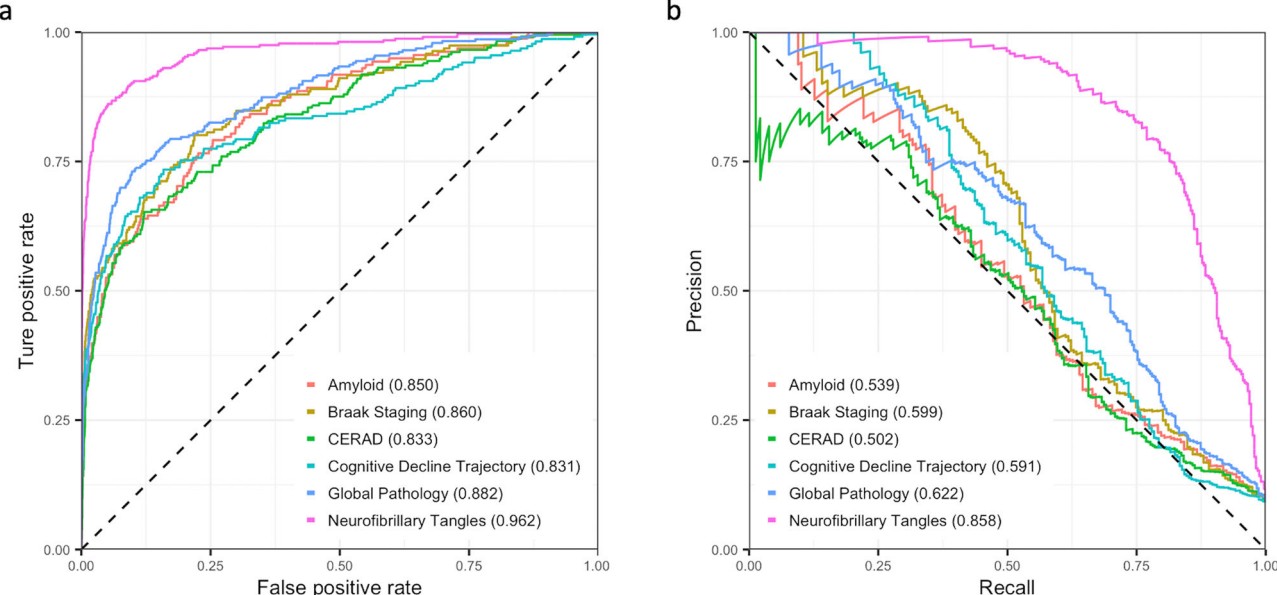

**Fig. 2 Summary EWASplus results. a** ROC curves of the predictive performance of EWASplus on the six traits in the ROS/MAP cohort. **b** Precision-recall curves of the predictive performance of EWASplus on the six traits in the ROS/MAP cohort. Source data are provided as a Source data file.

EWASplus prediction scores between suggestively positive and negative CpGs that are not in the training sets (Wilcoxon rank-sum test; $p < 3.64 \times 10^{-16}$; Supplementary Fig. 2) for all six traits. Scores for CpGs in group b are similar to those in group a (positive training set), albeit with higher variation, whereas scores in group c have almost the same scores as group d (negative training set). As expected, our results demonstrated excellent capability of EWASplus in distinguishing CpGs that show AD association or not.

**EWASPlus performance for off-array CpGs**. We applied the six classifiers trained on the six AD traits using EWASplus to the entire human genome to obtain a prediction score for every CpG (Fig. 3a). The top ten CpGs with the highest composite scores are listed in Table 2. The total number of CpGs with a prediction score is about 78 times the number of CpGs present on the Illumina 450K methylation array. The prediction scores for all CpGs are provided at the EWASplus Github site.

For the off-array CpGs, we examined the distribution of prediction scores for different types of genomic regions. We hypothesized that top CpGs with the highest prediction scores would be located in functional regions such as enhancers and promoters, and we find this to be the case (Fig. 3b). The normalized proportion for enhancers ranges from 15.73 to 43.19% and exons range from 4.69 to 23.99%, which are both significantly higher than the expected occurrence of these regions in the high prediction score percentile intervals (binomial test for the highest prediction score quantile interval: $p < 1.00 \times 10^{-99}$ for both enhancers and exons). To better understand the properties and context of top-ranked CpGs predicted by EWASplus, we selected the top 10k CpGs with the highest overall EWASplus prediction scores and analyzed their chromatin states (15-state model) defined in dorsolateral prefrontal cortex. We calculated the enrichment (or depletion) of the 15 chromatin states in the top 10k CpGs. As a result, we found that all six AD-related traits are enriched for sites annotated as flanking active transcription start site (TSS) (binomial test; $p < 1.00 \times 10^{-99}$ for all traits), active TSS (binomial test; $p < 1.00 \times 10^{-99}$ for all traits), enhancers (binomial test; $p < 1.22 \times 10^{-9}$ for all traits), and repressed polycomb (binomial test; $p < 1.00 \times 10^{-99}$ for all traits),

and under-represented for sites within quiescent regions (binomial test; $p < 1.00 \times 10^{-99}$ for all traits) (Fig. 3c). There is no significant difference in the enrichment patterns across the six AD traits. These results support the conclusion that top CpGs associated with AD tend to be located in functional regions.

**Comparison with a competing method**. In a recent work, using array-measured methylation levels, Zhang et al.[21] develop a computational algorithm to impute the methylation levels on CpG sites genome-wide including those not on the Illumina 450K array. Their approach employed about 125 genomic and epigenomic features (the number varies when including different sets of individual-level features) mainly composed of regulatory marks from ENCODE project. Although not designed for trait-association prediction, one could apply this method to impute methylation levels for every individual sample and on every CpG site. Subsequently, association test can be conducted on these imputed methylation measures to identify CpGs significantly associated with a trait of interest.

To compare such a strategy with EWASplus, we applied Zhang et al.'s method and used the imputed methylation values to conduct an association test. We found that the AUC for EWASplus is between 0.178 and 0.329 higher compared to the adapted Zhang et al. approach; AUPR for EWASplus is 0.219 to 0.364 higher than adapted Zhang et al. approach across six AD-related traits (see Supplementary Fig. 3a, b for performance comparison).

**Experimental validation of EWASplus predictions**. To experimentally test the validity of the prediction scores reported by EWASplus, we performed targeted bisulfite sequencing to measure the methylation level at 559 selected CpGs from 150 randomly selected participants from the Religious Orders Study (ROS) or Memory and Aging Project (MAP) cohorts who are representative of both studies and have available brain tissue for bisulfite sequencing (Supplementary Table 9). CpGs were selected for independent validation from the top EWASplus predicted sites using a stepwise selection process that prioritized regions with the highest predicted scores that were physically separated by at least 500 bp. For comparison purposes, we also randomly

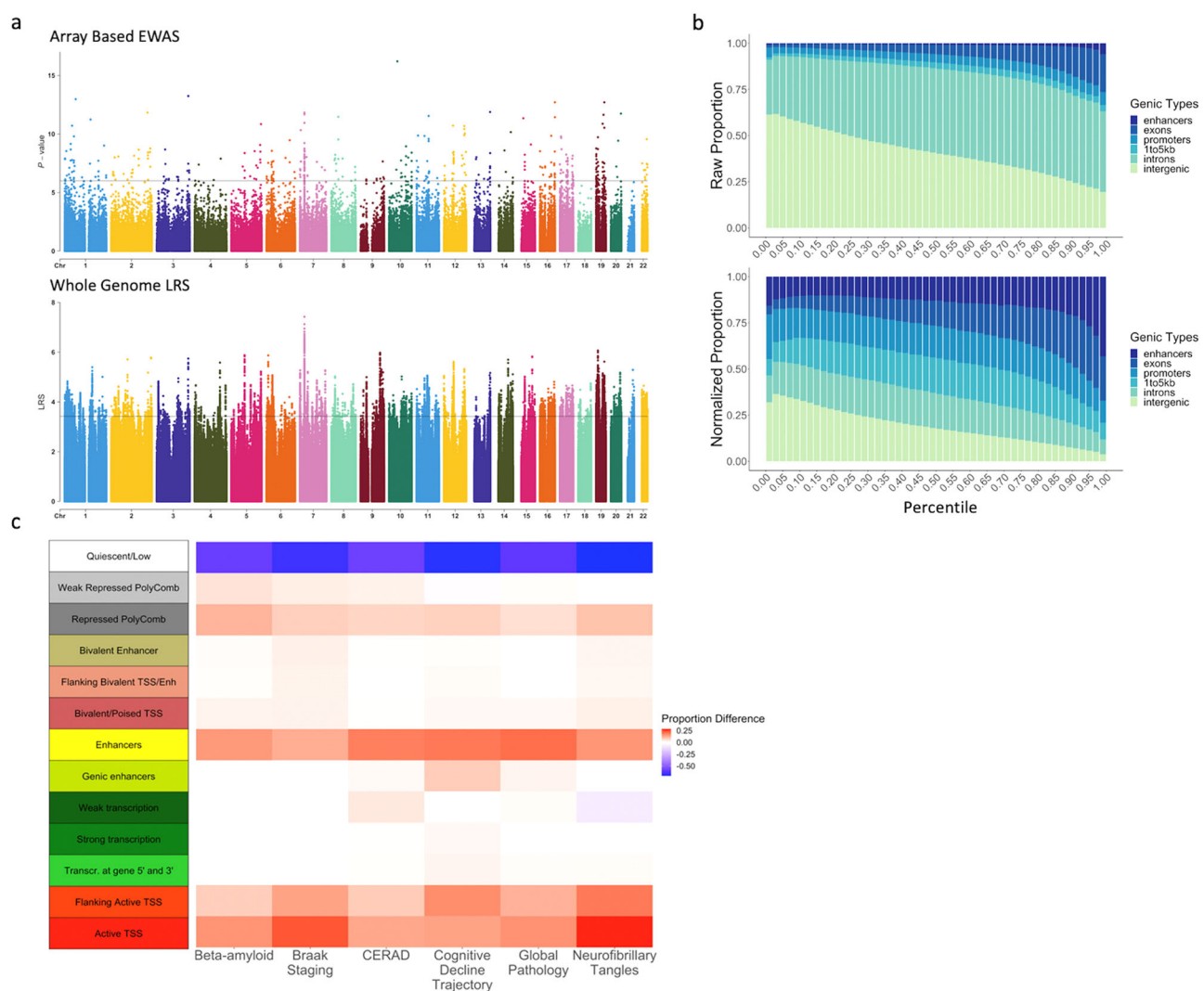

**Fig. 3 Genome-wide prediction results. a** Manhattan plots for neurofibrillary tangles: the top panel is for on-450K CpGs with EWAS $p$-values and the bottom panel is for whole-genome CpGs with imputed LRS by EWASplus. The $y$-axis is the log-scale rank scores. The top-ranked CpG has the LRS of 7.42 (about empirical $p$-value of $3.8 \times 10^{-8}$); the top 100th ranked CpG has the LRS of 5.42 (about empirical $p$-value of $3.8 \times 10^{-6}$) and the top 10,000th ranked CpG (about empirical $p$-value of $3.8 \times 10^{-4}$) has the LRS of 3.42. **b** Raw and normalized stacked-proportion histograms for different genomic annotation types. Source data are provided as a Source data file. **c** The difference of observed and expected chromatin states proportion for the top 10,000 loci across the six AD-related traits: Beta-amyloid, Braak staging, CERAD, cognitive trajectory, global pathology, and neurofibrillary tangles. Source data are provided as a Source data file. The annotated chromatin states are from Roadmap Epigenetics Project and we used the core 15-state model chromatin states for the dorsolateral prefrontal cortex tissue type. To minimize ambiguity, we require only a single annotation type is assigned for each CpG site. if a CpG has multiple annotations, we only record the most "significant" annotation with the following order: enhancer > promoter > exon > intron > near gene (1–5 kb to the TSS) > intergenic. We do not list 5' UTR and 3' UTR since these two types are within the first and last exon of each gene according to the UCSC annotation system.

selected CpGs from regions with predicted scores in the lower half but similar physical characteristics (e.g., GC content). In addition, we targeted CpGs on the array that could serve as positive controls. After quality control, 319 CpGs were analyzed including 31 CpGs on the 450K array identified as AD-associated[22], 260 off-array CpGs predicted to be AD-associated based on EWASplus, and 28 off-array CpGs predicted to not be AD-associated. These 319 CpGs can be grouped into 58 independent clusters (referred to as CpG cluster hereafter) on the genome that belongs to three groups: 38 off-array predicted AD-associated, 10 on-array AD-associated[22], and 10 off-array predicted not AD-associated. For performance comparison, we combined test results from the six individual traits. Due to the limited sample size, we call a CpG cluster AD-associated if at least one of the CpGs at the locus achieves unadjusted $p$-value for

differential DNAm <0.05 for any of the six traits. Similar to our results from individual traits, we found that positive CpG clusters predicted by EWASplus have the highest rate of association with at least one AD trait (65.8%, or 25 of 38), followed by CpG clusters identified by array-based EWAS (60.0%, or 6 of 10). In contrast, the negative control CpG clusters predicted by EWASplus have the lowest (30.0%, 3 of 10) (Table 3). Thus, CpGs with top EWASplus scores are about 2.2 times more likely to be associated with an AD trait (Binomial test, $p < 1.00 \times 10^{-9}$).

**EWASplus performance on multiple cohorts.** To further test EWASplus, we examined its performance using data from three additional cohorts: London cohort[17] (prefrontal cortex, $N = 113$), Mount Sinai cohort[23] (prefrontal cortex, $N = 146$), and Arizona

**Table 2 Top ten CpG loci for six AD-relevant traits.**

| Chr | Position (bp) | Beta-amyloid | Braak staging | CERAD | Cognitive decline | Global pathology | Neurofibrillary tangles | Genes within 50 kb of associated CpG |
|---|---|---|---|---|---|---|---|---|
| 7 | 27148225 | **5.605** | **5.037** | **3.799** | **4.495** | **4.612** | **7.424** | **HOTAIRM1, HOXA-AS2, HOXA2, HOXA-AS3, HOXA1, HOXA2, HOXA3, HOXA4, HOXA5, HOXA6, HOXA7** |
| 5 | 172175606 | 4.679 | 4.662 | 5.009 | 4.658 | 6.248 | 3.720 | DUSP1 |
| 7 | 47367933 | 4.008 | 4.736 | 5.251 | 4.254 | 5.278 | 5.210 | TNS3 |
| 19 | 46270392 | **4.462** | **4.447** | **6.311** | **4.362** | **4.937** | **3.727** | **FBXO46, SIX5, DMPK, DMWD, RSPH6A, SYMPK** |
| 6 | 35286078 | 5.141 | 5.947 | 4.497 | 3.731 | 3.762 | 5.072 | ZNF76, DEF6, PPARD |
| 19 | 10736075 | 4.130 | 5.977 | 3.387 | 4.047 | 4.400 | 5.845 | APIM2, SLC44A2, ILF3, ILF3-AS1 |
| 9 | 116225986 | 3.408 | 4.977 | 4.253 | 3.630 | 5.307 | 5.893 | C9orf43, RGS3 |
| 1 | 59280358 | 3.065 | 6.248 | 3.973 | 5.179 | 5.130 | 3.755 | LINC01135, JUN |
| 19 | 15563592 | **6.579** | **4.549** | **3.132** | **3.849** | **4.009** | **5.215** | **MIR1470, AKAP8L, WIZ, RASAL3, PGLYRP2** |
| 7 | 151433271 | 3.577 | 4.432 | 4.265 | 4.739 | 4.743 | 4.814 | PRKAG2 |

For each AD-associated trait, this table provides a Log-Scale Rank Scores (LRS) where an LRS of 1 corresponds to 90% percentile, LRS = 2 corresponds to 99% percentile, and so forth. Genes within 50 kbp of the region are provided. Genes with prior evidence of being associated with AD given in bold.

cohort[24] (middle temporal gyrus, $N = 302$). In all three studies, Braak staging (treated as a continuous variable) is used as the trait in the EWAS studies, as described in Smith et al.[25]. Detailed information about these cohorts is summarized in Supplementary Table 7.

We found that EWASplus performed well in all three datasets. The AUC values range from 0.697 (London 1) to 0.863 (Mount Sinai) (Supplementary Fig. 4a). The AUPRC values range from 0.233 (Arizona) to 0.604 (Mount Sinai) (Supplementary Fig. 4b). The complete results including all evaluation metrics can be found in Table 4.

To understand the most relevant factors influencing EWASplus performance among the different datasets, we treated the performance measurement testing AUC as the response variable and tested numerous independent variables using the linear regression model. We found that when choosing the positive EWAS threshold (negative logarithm transformed *p*-values) as the independent variable, simple linear regression achieved $R^2$ of 0.588 using other performance measures such as AUPRC and F1 values produced similar results (Supplementary Fig. 5). These results suggest that perhaps the most relevant factor that influences EWASplus performance is the quality and power of the original EWAS, which depends on the effect and sample sizes.

**Biological insights into AD**. To glean biological insights from the EWASplus results, we examined genes surrounding some of the highest EWASplus scoring CpGs. Interestingly, we found that the highest scoring CpG is located inside the *HOXA* gene cluster, which has been identified by three independent array-based EWASs of cortical brain tissue associated with Braak staging, a measure of neurofibrillary tangles[17,23,26]. In contrast to prior analyses that identified individual HOX genes, EWASplus results identify a 40 kb region on chromosome 7 that includes multiple homeobox genes, e.g., *HOXA2*, *HOXA3*, *HOXA4*, *HOXA5*, and *HOXA6*, that are associated with AD (Fig. 4c).

In addition, of the top 10 detached EWASplus scoring CpGs, seven were not previously implicated in any EWAS of AD. Here detached means any two CpGs on this list are at least 10 kb away from each other. Gene set enrichment analysis by GeNets[27] using all genes located within 5 kb of the top 100 EWASplus scoring CpGs (123 genes, Supplementary Data 1) revealed a significant enrichment of protein kinases ($p = 0.044$, Supplementary Fig. 6) —*ALPK3*, *DMPK*, *MAP3K11*, *MAP4K1*, and *TAOK3*[28]. Identification of kinases within AD is of particular interest given that neurofibrillary tangles, a hallmark neuropathology of AD, result from hyperphosphorylation of microtubule-associated protein tau (MAPT)[29]. In addition, we found that genes within the top EWASplus regions have evidence of physical interaction with known AD genes or AD GWAS loci (Supplementary Fig. 6) (e.g., *PRKAG2* and *TNS3* interact with *APOE*, *CLU*, *APP*, *PSEN1/2*, and *RIN2* and *RIN3* interact with *BIN1*). These analyses support the idea that EWASplus is able to identify interesting underlying biological relationships in AD.

**Discussion**
EWAS has been shown to be a powerful and effective approach to derive associations between methylation changes and phenotypes. EWAS studies of human brain have elucidated additional genes involved in AD[16–19]. To expand our understanding of potential AD-relevant regions in the genome, we developed EWASplus to explore the 97% of CpGs that are not included on the methylation arrays. EWASplus uses an ensemble learning-based computational pipeline to learn relevant features from a large set of potential omics features.

**Table 3 Comparison of number and proportion of differentially methylated CpGs in various categories of CpGs.**

|  | # of positives in EWASplus predicted positives (%) total = 38 | # of positive in on-array positives (%) total = 10 | # of positives in EWASplus predicted negatives (%) total = 10 |
| --- | --- | --- | --- |
| Any trait | 25 (65.8) | 6 (60.0) | 3 (10.0) |
| Beta-amyloid | 17 (44.7) | 1 (10.0) | 2 (20.0) |
| Braak staging | 11 (28.9) | 2 (20.0) | 1 (10.0) |
| CERAD | 17 (44.7) | 2 (20.0) | 3 (30.0) |
| Cognitive trajectory | 7 (18.4) | 3 (30.0) | 0 (0.0) |
| Global pathology | 13 (34.2) | 2 (20.0) | 3 (30.0) |
| Neurofibrillary tangles | 16 (42.1) | 5 (50.0) | 1 (10.0) |

Methylation level is measured by targeted bisulfite sequencing experiment.

**Table 4 Summary of performance evaluation on three additional cohorts of samples: London, Mount Sinai, and Arizona.**

| Cohort | Brain tissue | AUC | AUPR | F1 | Precision | Recall |
| --- | --- | --- | --- | --- | --- | --- |
| London | Prefrontal cortex | 0.697 | 0.272 | 0.325 | 0.248 | 0.471 |
| Mount Sinai | Prefrontal cortex | 0.863 | 0.604 | 0.481 | 0.364 | 0.708 |
| Arizona | Middle temporal gyrus | 0.699 | 0.233 | 0.275 | 0.196 | 0.461 |

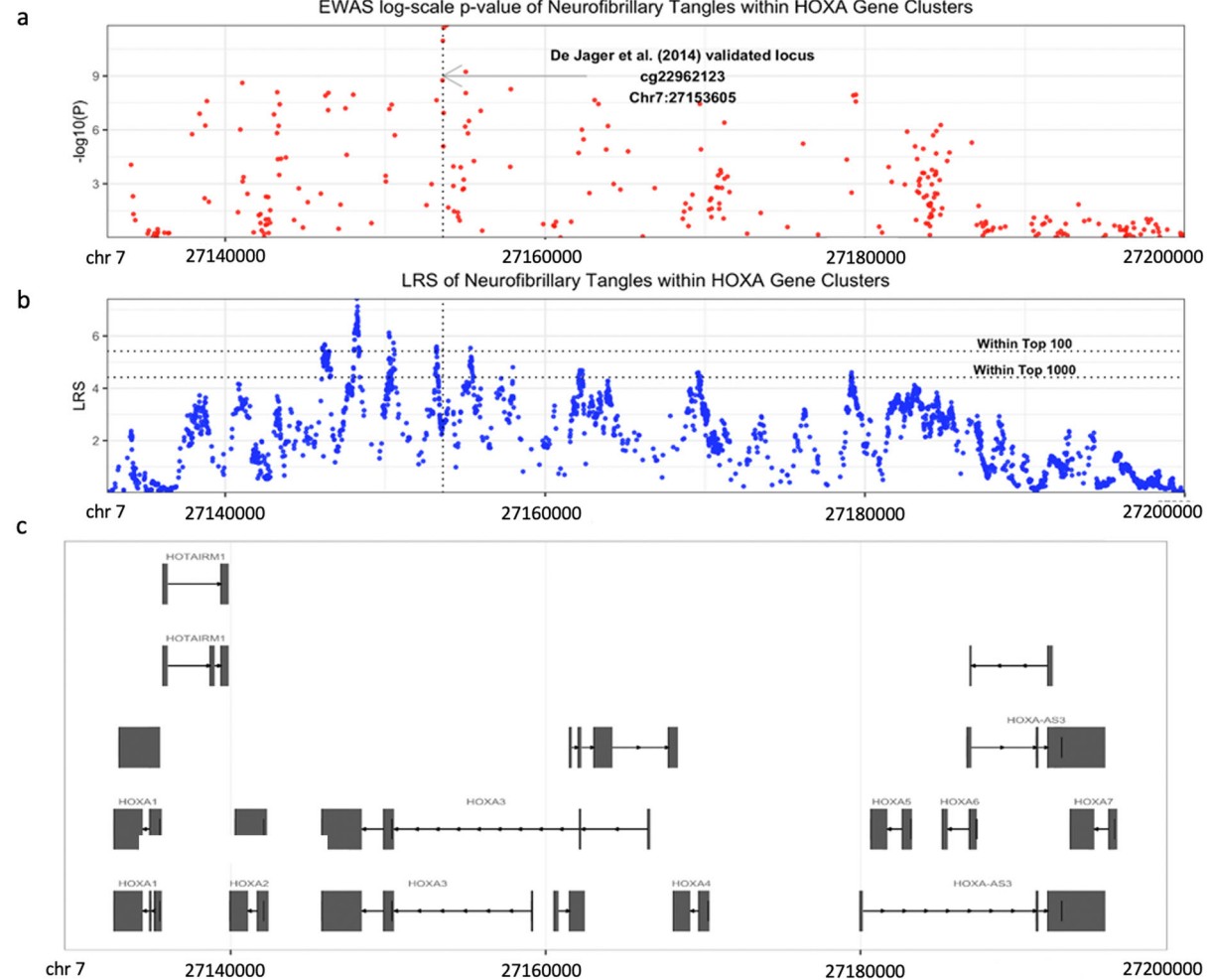

**Fig. 4 Manhattan plot of neurofibrillary tangles EWAS at the HoxA locus on chromosome 7. a** Array-based EWAS *p*-values. The most significant CpG identified by De Jager et al. are shown with an arrow. **b** EWASplus predicted LRS. **c** The landscape of the HoxA cluster genes.

EWASplus is a powerful machine learning method based on disease-specific EWAS results and has some parallels with genotyping imputation strategy used in genetics studies[30]. The fundamental difference is that genomic imputation relies on linkage disequilibrium[31], but DNAm does not share the same degree of physical correlation[21]. In fact, the correlation of methylation levels between two adjacent CpGs decays rapidly with distance[32,33]. Thus, EWASplus takes an alternative approach by inferring whether a CpG is trait-associated. This is achieved under a supervised classification framework.

EWASplus can effectively identify AD-associated differentially methylated CpGs according to multiple experiments conducted to evaluate its performance. First, using only CpGs on the methylation array, in silico cross-validation revealed high AUC and AUPRC for all six traits. Second, we observed good separation of EWASplus prediction scores between near positive CpGs versus negative CpGs not in the training set (Supplementary Fig. 2; Wilcoxon rank-sum test $p$-value ranges from $2.82 \times 10^{-99}$ to $3.64 \times 10^{-16}$) in EWASplus prediction scores. For sites not assayed by the methylation array, we found significant enrichment of high-scoring CpGs in genomic regions of functional annotations such as TSS regions and enhancers. Finally, and most importantly, we performed experimental validation using targeted bisulfite sequencing on CpGs not included on the methylation arrays.

Our EWASplus results are notable, in general, for two reasons. First, high-scoring EWASplus CpGs are more likely to be located in regions with functional annotations such as enhancers or promoters. Both of these results are consistent with other work showing that gene regulation is a key facet of many diseases[34,35]. Second, EWASplus results illustrate how epigenetic "fine mapping" may illuminate disease pathophysiology. For example, in the *HOXA* locus EWASplus results suggest that epigenetic changes are occurring across the gene cluster in AD rather than one gene-family member.

A key idea of EWASplus is that it bypasses inferring the individual-level DNAm level directly. A similar approach has been used to predict additional trait-associated genetic variants using GWAS and machine learning[36]. Since our goal is to identify disease-associated DNAm CpGs rather than methylation status directly our approach avoids much complexity associated with accounting for the many factors that can influence DNAm CpG status (e.g., age, cell type proportion). This is illustrated by the performance of EWASplus compared to the modification of Zhang et al.'s method to address disease-association, which it was not originally designed to do, admittedly.

EWASplus results for AD reveal several interesting biological insights. First, we identified a 40 kb region in the homeobox A cluster of genes that are associated with AD, which expands upon the previously described association with individual genes within that cluster (e.g., *HOXA3*) and AD. Since these are known transcription factors, these findings may suggest important transcriptional regulation occurs in AD or its progression. Second, we find enrichment of kinases—*ALPK3*, *DMPK*, *MAP3K11*, *MAP4K1*, and *TAOK3*—in the top EWASplus loci. This finding is particularly relevant for AD given that the pathologic hyperphosphorylation of tau is a hallmark neuropathologic feature of AD (i.e., neurofibrillary tangles). Of these kinases, only *ALPK3* and *MAP4K1* were previously suggested to associate with AD[23,37–39]. *DMPK* is notable for causing myotonic dystrophy type 1 due to a repeat expansion within an intronic region in carriers that leads to altered gene expression of genes within that region[40]. Interestingly, differential DNAm of *MAP4K1* has been associated with AD in human hippocampus[39] and Braak staging (a measure of neurofibrillary tangle pathology)[23] in independent human brain datasets. TAOKs (thousand and one amino acid kinases, also referred to as prostate-derived STE20-like kinases [PSKs]) have been extensively investigated for their ability to phosphorylate MAPT and regulate microtubule assembly[41]; yet, to our knowledge, methylation of TAOK3 has not been previously associated with AD. Finally, from the top 10 EWASplus results (Table 2) we found four genes that have intriguing connections with AD or cognitive decline from approaches other than methylation. These genes include *DUSP1*, *PPARD*, *JUN*, and *PRKAG2*. For example, a *PPARD* null mouse model shows cognitive impairment[42], and *PPARD* is highly expressed in the brain[43] and implicated in type 2 diabetes and obesity[44], which are risk factors for AD. In addition, there is experimental evidence to suggest that *JUN* and *PRKAG2* regulate or interact with *APP*[45,46], which is of interest in AD given *APP* is cleaved to beta-amyloid. Thus, these findings from the literature provide complementary support that EWASplus identifies disease-relevant findings and is likely to provide fresh insight into AD.

DNA methylation is tissue-specific. Most of the tests done in this study are conducted on the PFC region of the brain. We focused on PFC for several reasons. First, epigenetic marks are correlated across neocortical regions[47]. Second, cell loss in PFC is relatively less even in people with high neuropathological burden from AD compared to other cortical regions. Third, the majority of available reference human brain transcriptomes and proteomes are from the PFC allowing future work to test predictions of EWASplus using existing data. Despite focusing on PFC, EWASplus performs well on the middle temporal gyrus. Thus, we expect EWASplus to perform well for other tissues because the genome-wide features used are from many different tissue types. From all the tests we performed, we found that the number and level of significant CpGs seem to have a strong impact on the EWASplus performance. Therefore, we are confident that EWASplus will be able to successfully extend the coverage of high quality, well-powered array-based EWAS studies.

Although the EWASplus methodology is general and can be applied to any tissue type, the methylation profiles are tissue-specific, may change with age/environment and demographics. This implies that the trained EWASplus model is only valid for the specific tissue type collected from samples with certain age/environmental profile and demographics. One should exercise caution when trying to extrapolate the results to other tissue types such as blood, or subjects with different age or environmental and demographic profiles. Since the major utility of EWASplus is to expand the coverage of EWAS beyond the array within a specific experimental dataset, this limitation will not hamper the utility of EWASplus.

A potential limitation of EWASplus is the limited number of underlying training datasets and the focus on subjects of recent European descent. Thus, it is of particular importance to expand the number and diversity of additional EWAS data in future work. The underlying methylation data were also from PFC, which is affected relatively late in AD; however, the findings may not generalize to other neocortical regions. Thus, training data from additional relevant brain regions would improve EWASplus models. Likewise, while we started with a large number of potential features, many were from non-neuronal sources, which may limit generalizability to brain tissue. However, as those data are generated, our approach can be easily retrained with those data for improved specificity for brain-cell types from different regions. A strength of this work is that the underlying methylation data were derived from participants enrolled in a population-based study of aging, and there is a wide range of neuropathology findings that reflects the general population rather than a clinic-based ascertainment[48]. We also show a high degree of

experimental validation and note that future work could employ targeted bisulfite sequencing[49] or a custom array platform[50] to profile candidate CpGs in a cost-effective and high-throughput manner.

EWASplus does not provide a significant cut-off threshold since it is a supervised classification approach, not a testing-based method. In practice, one can select the threshold empirically by checking whether top CpGs identified by array-based EWAS made the cutoff. Deciding on the number of significant EWAS CpGs to include in training is a tradeoff between the quantity and quality of the training set in EWASplus. Thus, the significance threshold for each EWAS should be decided based on the effect size and sample size of the EWAS. Future work should examine the utility of including different thresholds and use cross-validation to select the desired significance cutoff. For a CpG, no matter how highly ranked by EWASplus, should only be considered as "putative" in terms of trait association unless it can be validated using experimental approaches.

In conclusion, we present EWASplus, a powerful machine learning approach to identify disease-associated CpGs with high reliability. Application of EWASplus to AD highlights important regions and genes that likely contribute to AD pathogenesis, which a valuable addition to the investigation of the epigenetic landscape of AD. In addition, EWASplus is a general approach that may be applied to extend any existing EWAS results obtained using array-based technology, regardless of the trait or phenotypes being studied. We anticipate more exciting findings from its future applications.

## Methods

**Cohorts**. The main dataset used in this study comes from the ROS/MAP cohorts. ROS and MAP are longitudinal cohort studies of aging and AD led by investigators at the Rush Alzheimer's Disease Center[51,52]. Participants give written informed consent for annual assessments, signed an Anatomic Gift Act, and a repository consent to allow their data and biospecimens to be repurposed. Each year, participants undergo a detailed medical, neurological, and neuropsychiatric assessment. After death, each participant undergoes a detailed brain autopsy with neuropathologic examination. Both ROS and MAP were approved by the Institutional Review Board of Rush University Medical Center. They share a large common core of data at the item level to allow efficient merging of datasets. ROS/MAP resources can be requested at https://www.radc.rush.edu.

In addition, we also obtained data from three separate cohorts: London, Mount Sinai, and Arizona. The "London" cohort refers to prefrontal cortex tissue obtained from 113 individuals archived in the MRC London Neurodegenerative Disease Brain Bank. The details of the cohort are described in Lunnon et al.[17]. The "Mount Sinai" cohort refers to prefrontal cortex tissue obtained from 146 individuals archived in the Mount Sinai Alzheimer's Disease and Schizophrenia Brain Bank. Details of this cohort is described in Smith and colleagues[25]. The "Arizona" cohort refers to 302 middle temporal gyrus samples from The Sun Health Research Institute Brain Donation Program[24]. The details of this cohort are described in Brokaw et al.[53].

### Sample preparation and differential DNAm CpGs identification. DNAm data were generated from dorsolateral PFC (Broadman area 46) of post-mortem samples obtained from individuals in the ROS/MAP cohorts.

DNAm profiling was performed with the Illumina HumanMethylation450 Beadchip array[16]. After excluding non-Caucasian subjects, 717 ROS/MAP participants with array DNAm data remained for analysis. We obtained raw IDAT files from the Synapse website (Synapse ID: syn7357283) and removed probes annotated to multiple chromosomes or the X and Y chromosomes by Illumina, probes that cross-hybridize with other probes due to sequence similarity (identified by Chen et al.[54]), probes with a detection $p$-value > 0.01 in any sample, probes without a CpG, and probes that overlap with a common SNP (identified by Barfield et al.[55]). After this filtering, a total of 334,465 autosomal CpGs remained for analysis.

For the EWAS analyses, each probe was normalized using the BMIQ algorithm from the Watermelon R package[56], and adjusted for batch effects using the ComBat function from the sva R package[57]. We used the CpGassoc[58] R package to test if the methylation level of each array CpG is associated with the trait of interest via regression methods[58]. All models were adjusted for proportion of neurons, age at death, sex, post-mortem interval, plate, study, and years of education. Neurons were added as a covariate to avoid potential confounding due to differences in the

cellular composition of the tissue samples. The proportion of neurons in each sample was estimated using the CETS R package and reference methylation data from isolated neuronal nuclei[59].

We performed EWASs for the following six AD-related traits: (1) beta-amyloid load which is the percent area of beta-amyloid based on image analysis; (2) neurofibrillary tangle density by stereology; (3) CERAD score; (4) Braak stage; (5) global AD pathology burden; (6) cognitive trajectory based on the average z-score of 17 cognitive function tests. Beta-amyloid and neurofibrillary tangle were measured in the cortex using immunohistochemistry with antibodies specific to beta-amyloid and phosphorylated-tau, as described[52]. We used square-root-transformed values for both traits to improve their normality. CERAD score and Braak stage are semi-quantitative measures that reflect both a neuropathologist's opinion of AD diagnosis and the distribution and amount of silver-stain-identified neuritic and diffuse plaque and neurofibrillary tangle pathologies, respectively[60–62]. CERAD scores can take on values from one to four indicating definite AD, probable AD, possible AD, and no AD, respectively. CERAD was treated as a continuous trait. Braak stages can take on values from one to six, indicating the increasing spread of neurofibrillary tangle pathology in the brain, and Braak was coded as a binary trait with stages one to three as controls and stages four to six as affected. Global AD pathology burden is a summary measure of silver-stain-identified neuritic plaque, diffuse plaque, and neurofibrillary tangle pathologies[52]. As global AD pathology burden has a skewed distribution, we used square-root-transformed values. Cognitive trajectory, or the rate of change in cognition over time, was estimated for each ROS/MAP participant using a linear mixed model[22]. For each person, cognitive trajectory was estimated as the person-specific random slope of a linear mixed model that included global cognitive function as the longitudinal outcome[63], follow-up year as the independent variable, and sex, age at enrollment, and years of education as covariates.

For the London, Mount Sinai, and Arizona cohorts, we directly used the processed EWAS results reported in Smith et al.[25]. Details of the sample preparation and differential DNAm CpGs identification have been described in previous studies[17,24,53].

### Training sets selection. For each trait, positive CpGs in the training set were selected based on association test $p$-values (threshold ranges from $1.00 \times 10^{-7}$ to $1.00 \times 10^{-5}$). For each positive CpG, ten matching negative CpGs were selected from the Illumina 450K array such that they have similar β-values as the positive CpG, but none is considered significant in any of the EWASs conducted on the six traits. We used a conservative threshold ($p > 0.40$) for being not-significant, and β-values were calculated as the mean values of methylation intensity over 717 ROS/MAP samples for each CpG on the Illumina 450K array.

### Base classifiers. We used four different methods as base classifiers with varying model complexity. The goal of this approach was to select the model with the least error to achieve an optimal overall performance. We used four models that included: (1) RLR with L2-penalty, which alleviates overfitting and feature collinearity; (2) SVM classifier[64], which performs well with linearly non-separable classification, a common feature for real-world problems; (3) RF[65], which is a bagging method with decision tree as base learner; (4) GBDT[66], which differs from RF in that a new tree is added to model to gradually optimize the objective function that was set as log loss. EWASplus uses an accurate and efficient implementation of GBDT from package XGBoost[67].

### Feature selection. We assembled a comprehensive collection of 2256 genomic/epigenomic profiles as well as multiple functional annotation scores as features to be used in the model. Omics profiles include TF and histone ChIP-seq, open chromatin, total RNA-seq, and WGBS. Functional annotation scores include CADD[68], GenoCanyon[69], and Eigen/EigenPC[70]. More details can be found in Supplementary Table 10.

The moderate size of the training sets (between 1706 and 3181) may result in overfitting if all features are included in the training. Thus, we used a dimension reduction/feature selection step before the model training. For each trait, we performed feature selection for each of the four base classifiers: RLR, SVM, RF, and GBDT, respectively. For each base classifier, we selected the top 100 most informative features using the training data. In RLR and SVM, features were ranked based on the weights of the fitted model. For RF, features were ranked based on the Gini impurity measure. For GBDT, features were ranked by the gain metric when fitting the model, or, in other words, the improvement in accuracy brought by a feature to the branches it is on.

Next, we ranked the features by the number of times that this feature was selected by the four base learners as informative. We selected the top 60 features (testing on the number of top features ranges 30–100, 60 was selected because it gave the best performance overall). Features were ranked by the number of methods that select the feature as informative. To break a tie, we introduced a secondary sorting method. For each feature, we conducted the Wilcoxon rank-sum test comparing feature values between positive and negative CpGs, and features were ranked from the most significant to the least significant. The top 60 features for all six traits from the ROS/MAP cohort and Braak staging from three additional

cohorts are shown in Supplementary Data 2–10. These features may be informative about underlying disease mechanisms.

**Hyperparameter tuning and ensemble model**. We used the Tree-of-Parzan Estimators (TPE) implemented in Hyperopt[71] to adaptively search the hyperparameter space of each component model (base learner) for the best hyperparameter settings. This model-based hyperparameter tuning method is thought to achieve better performance than random search in terms of both accuracy and efficiency[71].

The hyperparameter tuning for each component model is conducted separately. In the training dataset, we uniformly up-sample the positive CpGs to match the number of negative CpGs to alleviate the imbalance problem. In the outer CV, the whole dataset of positive and negative CpGs were split into training and testing sets in a nine-to-one ratio in each round. Within each round, the ninefolds were further split into threefold to conduct the inner 3-fold CV for hyperparameter searching. The best set of hyperparameters was decided by the highest F1 score and it was then used for the remaining onefold in the outer 10-fold CV. Each one of the tenfolds in the outer CV layer is used once as the testing set in a round-robin way so that out-sample predictions cover the whole dataset. We evaluate our model with the out-of-bag estimates for testing error and report the evaluation results in Table 1 for the ROS/MAP cohort and Table 4 for other additional cohorts.

After the outer 10-fold CV, we then built the ensemble model by selecting the best combination of component models. The out-sample predictions of each base learner from the outer 10-fold CV were aggregated in a soft-voting manner to give the ensemble prediction probabilities in different combinations of component models. Due to the problem of class imbalance, we evaluated the performance of the ensemble models using AUC, AUPR, precision, recall, and F1 score.

**Performance evaluation metrics**. To assess the performance of EWASplus, we used three classes of evaluation metrics: precision and accuracy, AUC and AUPRC, as well as F1 score. Precision measures the true positive rate of a classifier. Accuracy measures the percentage that a classifier correctly labels test samples. For imbalanced datasets where positive samples are of more interest, precision is preferred over accuracy. PRC is preferred over ROC. The F1 score is another widely used performance measure for imbalanced datasets. It takes into consideration both accuracy and precision by assigning each an equal weight in the following calculation formula: $F_1 = 2 \times \left( \frac{recall*precision}{recall+precision} \right)$. The focus of F1 score is on the positive samples which is usually under-represented.

**Binomial test for enrichment of protein kinases**. We selected the top 100 CpGs with the highest EWASplus prediction scores across six AD-related traits in a stepwise forward manner such that any two CpGs in the top 100 list are at least 10 kb away from each other. Next, we searched through the 5 kb neighborhood of these 100 CpGs to retrieve all genes that overlapped, for a total of 123 genes. Among these genes, five are known protein kinases. Given a complete list of human kinases (492 from the autosomes) from Kinase.com (http://kinase.com/human/kinome/)[28] and a complete list of human genes (31,684 from the autosomes) from Ensembl (http://grch37.ensembl.org), we conducted an enrichment test using binomial distribution which returned an enrichment p-value of 0.044.

**Log-scale rank score (LRS) for prioritizing AD-associated loci**. In order to better present the whole-genome prediction result, we sorted the prediction scores of each trait and calculated the log-scale rank score (LRS) for each CpG (LRS $= -\log_{10} \frac{rank}{total\ num\ CpGs}$; total num CpGs $= 26,573,858$). The LRS is similar to a log-transformed empirical p-value. A higher LRS means the CpG is more likely to be associated with the trait.

**Loci selection for targeted bisulfite sequencing**. Targeted bisulfite sequencing was conducted on selected CpGs (with neighboring CpGs profiled unintentionally, as well) for 150 randomly selected samples from the ROS/MAP cohort. Since most features used in model training having only one value in every 200 bp bin, CpGs within a 200 bp bin tend to have similar prediction scores. In order to select a more representative (less clustered) set of loci for experimental validation, we required any pair of selected CpGs must be at least 500 bp apart. The forward selection process is performed in the stepwise manner, starting from the CpG with the highest total LRS score. Due to the limitation of sequencing primer design, not all loci on the candidate list were selected for bisulfite sequencing. The selection process was stopped when a pre-determined sequencing capacity is reached. For comparison, we selected 38 off-array CpG clusters with high prediction scores, 10 clusters of on-array CpGs listed in de Jager et al.[16] and 10 clusters of off-target negative control CpGs.

**Adaptation of Zhang et al. for comparison with EWASplus**. For the purpose of fair comparison, we selected 1000 CpGs that are not from the training set used by EWASplus. Instead, we selected 500 "near positive" CpGs with p-values just above

the threshold and 500 negative CpGs with p-values > 0.40 but not in the negative training set used by EWASplus. Comparison is performed in two steps: (1) predict methylation levels for the 1000 CpGs across the 717 samples used to train EWASplus following instruction in Zhang et al., and (2) perform association test with R package CpGassoc[58] to test for differential methylated based on the predicted methylation level from the first step.

**Targeted bisulfite sequencing**. Multiplex primers were designed to amplify the identified regions using MPD software[72] (primers list can be found in Supplementary Data 11). The 200–500 ng purified genomic DNA was used for bisulfite conversion (EpiTect Bisulfite Kit (Qiagen)). The treated DNA were used for PCR amplification and PCR amplicons were further purified and pooled together in equal molar. Mixed amplicons were then purified for libraries preparation and deep sequencing (100× or above) using a MiSeq following standard procedures recommended by Illumina. Image analysis and base calling were performed using standard Illumina pipelines. Quality control was performed in the same fashion for the array-based genotyping, except the missingness threshold was raised to 50%.

Association testing was performed using the same approach as the array-based methylation with CpGassoc[58] with modifications. The methylation levels were modeled as logit transformation of $\beta$ values ($\log(\beta/1 - \beta)$) to stabilize the variance[73]. Next, we grouped adjacent CpGs into clusters and conducted the test for differential DNAm. Due to the limited sample size of the study, we call a CpG cluster differential DNAm if the lowest p-value from the Differential DNAm test is less than an unadjusted p-value 0.05 among all CpGs in the CpG cluster. We adjusted for the following covariates: age of death, sex, years of formal education, post-mortem interval, study, and cell type proportion, which was estimated using the CETS R package[59].

**Protein–protein interaction and pathway analyses**. To identify potential crosstalk among known AD genes and genes suggested by EWASplus, we used web platform GeNets[27] (https://apps.broadinstitute.org/genets) to query a combined list of 28 known AD-associated genes and 123 genes near the top 100 detached CpGs ranked by EWASplus prediction scores (Supplementary Data 1).

**Reporting summary**. Further information on research design is available in the Nature Research Reporting Summary linked to this article.

## Data availability

EWASplus annotated genome-wide risk scores for six AD-related traits, the processed features used for EWASplus training, the list of gene names residing in the flanking regions of the top predicted CpG loci, and the result of protein–protein interaction and pathway analyses are available at https://figshare.com/collections/Dataset_collection_of_EWASplus/5430207 (https://doi.org/10.6084/m9.figshare.c.5430207). The data that support the findings of this study (The processed EWAS data from the London, Mount Sinai, and Arizona cohorts) are made available to us from the authors of Smith et al.[25], but restrictions apply to the availability of these data, which were used under agreement for the current study, and so are not publicly available. Targeted bisulfite sequencing data and ROS/MAP EWAS summary statistics of all traits are available through Synapse (https://www.synapse.org/#!Synapse:syn25832093). Source data are provided with this paper.

## Code availability

The EWASplus software implements feature selection, ensemble model training, and annotation of risk-prediction scores. The source code is available via Github at https://github.com/xsun28/EWASplus/tree/remastered (https://doi.org/10.5281/zenodo.4770198).

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

## Acknowledgements

We are grateful to the participants and staff of the ROS and MAP for their time and participation. We are indebted to Ehsan Pishva, Danielle Brokaw, Katie Lunnon, Diego Mastroeni as well as the Lunnon group and the Mastroeni group for sharing EWAS summary data from the London, Mount Sinai, and Arizona cohorts. We thank Mesa Schumacher for graphical design help. We thank Leo Zhang for technical assistance. The work reported in this publication was supported in part by Imagine, Innovate and Impact (I3) Funds from Emory University School of Medicine, the Georgia CTSA NIH Award (UL1-TR002378), NIH awards R56 AG062256, R56 AG060757, R01 AG056533, the Accelerating Medicine Partnership for AD (U01 046152; U01 AG046161; U01 AG061356; U01 AG061357), the Emory Alzheimer's Disease Research Center (P50 AG025688), the NINDS Emory Neuroscience Core (P30 NS055077), Rush University Alzheimer's Disease Center (P30 AG10161), and grants supporting ROS/MAP studies (R01 AG017917, R01 AG015819, RC2 AG036547, RF1 AG036042, U01 AG61356). The views expressed in this work do not necessarily represent the views of the Veterans Administration or the United States Government.

## Author contributions

X.S., Y.H., Z.S.Q., T.S.W., and P.J. conceptualized and designed the study. X.S., Y.H., R.L., and Z.M. acquired the data. Y.H., X.S., H.J., S.Y., C.R., M.A., X.S., and E.S.G. conducted analyses. Y.H., X.S., Z.S.Q., T.S.W., and P.J. interpreted the data. X.S. and Y.H. wrote the first draft of the manuscript. X.S., T.S.W., Z.S.Q., P.J., A.P.W., D.A.B., and P.L.D.G. supervised the study. All authors critically revised and reviewed the manuscript.

## Competing interests

The authors declare no competing interests.
