## [Peer Review File · Nature Communications]

REVIEWER COMMENTS

Reviewer #1 (Remarks to the Author):

This is an interesting methodological paper that aims to allow the exploration of significantly more CpG sites than currently covered on the Illumina 450K arrays. The method is tested in Alzheimer's disease datasets, but will have utility for other disease-associated datasets and has the potential to be highly cited in the field. I had some specific comments, which are listed individually below:

1. In the introduction the manuscript discusses the various AD EWAS performed to date using the 450K array. It would be beneficial in the EWASplus Overview section of the results to say which cohorts were used and the number of samples for the training datasets
2. In the targeted bisulfite sequencing aspect of the methods it states "Due to the limited sample size of the study, we call a CpG cluster differential DNAm if the lowest p-value from the Differential DNAm test is less than an unadjusted p-value 0.05 among all CpGs in the CpG cluster." I was unclear what was meant by this, do they mean that provided any CpG in the cluster was nominally significant that then the cluster was deemed to be significant?
3. Figure 1 is low resolution and difficult to interpret, particularly for a non-expert. This figure should be improved explaining in more detail the steps taken and the outcomes.
4. Figure 2A is a table so should be shown as a separate table, rather than a component of figure 2.
5. Abbreviations in supplementary table 2A are not given. A more general reader will not know what the models are by the abbreviations. Abbreviations are also not shown in other supplementary tables too and legends could be extended for clarification.
6. Page 6 line 152 – the p value for exons is not properly given.
7. In the results the authors state that the best performance among the six traits for EWASplus was for neurofibrillary tangles. Why do they think this is?
8. The authors highlight in the discussion that a potential limitation of EWASplus is the limited number of underlying training datasets. As highlighted in the introduction there are several EWAS in AD using the 450K array aside from the ROSMAP data. It would be good to see the authors test their approach using these other datasets, many of which are publicly available. This would be important to test in other brain regions too
9. Have the authors considered using the approach with EPIC array datasets? In addition could the approach be validated using publicly available WGBS datasets with matched 450K data?
10. Given that methylation profiles are tissue specific and also can change with age/environment. Could the authors comment in the discussion about the application of the method and/or results to other tissues (such as blood) and in samples with different demographics

Reviewer #2 (Remarks to the Author):

DNA methylation is one of the most widely studied epigenetic changes. However, the Illumina 450K array only covers 2-3% of all CpG sites in the human genome and has known technical limitations. To fill this gap, Huang et al. proposed a novel machine learning method called EWASplus to predict the associated CpG sites at the whole genome level. Briefly, they use the soft-voting method to combine the results from four base learners (including penalized regression, support vector classifier, random forest, and XGboost). They applied EWASplus to six Alzheimer's disease (AD) related traits and validated parts of their findings experimentally. I have several comments/questions and hope the authors can consider addressing them.

1. There are several methods have been proposed to predict genome-wide DNA methylation. For example, Zhang, W. et al. *Genome Biology* 16, 14 (2015) proposed using the random forest to predict missing DNA methylation levels for all CpG sites. This seems to be highly related to the EWASplus, although they achieve the goal in a different way. The method proposed here (EWASplus) tries to divide the CpG sites into positive (i.e., signal) and negative CpG sites first and then predict the status by an ensemble method, while the method proposed in Zhang et al. tries to predict DNA methylation levels only. The author may consider comparing their method with these existing methods.
2. EWASplus directly predicts if a CpG site is associated with the trait of interest by using several features as predictors. This may induce one problem that the associations may go simply through those features. While EWAS is an association test and never imply causal, this may still count as a weakness of the EWASplus. In contrast, predicting DNA methylation levels directly and then conducting the association tests may reduce the concerns. This also backs to my Comment 2; the author may need to justify why they want to predict positive and negative CpG sites directly.
3. The method descriptions are not very clear. For example, it is unclear how to predict the missing DNA methylation levels based on the features listed in Supplementary Table 6. The feature names in Supplementary Table 6 are not very informative. Also, do you use both cis and trans information or only use the cis information only? Another example is the soft voting method. Do you use equal weights for each base learner or taking the base learner performance into account? Without these details, it is hard to evaluate the validity of the EWASplus.
4. DNA methylation levels will change depends on smoking status. Many studies incorporate smoking status (or predicted smoking status) as a covariate to reduce the concern of false positives. The authors may consider report genomic control factors.
5. The evaluation is based on cross-validation, which may be over-optimistic. If possible, using independent data or reserve an independent set to evaluate the performance may helpful.
6. Based on Figure 3A, it seems Array-based EWAS has much better performance than the proposed EWASplus. This is somewhat contradicted to Figure 2, which shows the AUC of Neurofibrillary Tangles is 0.962. Also, as EWASplus tests the whole genome's CpG sites, using the same cutoff as before may be problematic as they test more.

Response to reviews for NCOMMS-20-37924

Reviewer #1 (Remarks to the Author)

This is an interesting methodological paper that aims to allow the exploration of significantly more CpG sites than currently covered on the Illumina 450K arrays. The method is tested in Alzheimer's disease datasets, but will have utility for other disease-associated datasets and has the potential to be highly cited in the field. I had some specific comments, which are listed individually below:

1. In the introduction the manuscript discusses the various AD EWAS performed to date using the 450K array. It would be beneficial in the EWASplus Overview section of the results to say which cohorts were used and the number of samples for the training datasets

This is a great suggestion. We added a paragraph at the end of the EWASplus Overview subsection of the results section to list all the cohorts used in this study (page 5):

“EWASplus can be applied to any array-based EWAS to extend its coverage. In this study, we tested EWASplus on data collected from four different cohorts: ROS/MAP (sample size 717), London (sample size 113), Mount Sinai (sample size 146) and Arizona (sample size 302). Cohort characteristics are given in Supplementary Table 7 All original EWASs were performed using the Illumina 450K methylation array.”

Supplementary Table 7: Cohort Characteristics

	Braak Stage	N	Gender (M/F)	Age of death
London (N=113)	0-II	29	13/16	77.6(12.8)
	III-IV	18	7/11	88.5(5.2)
	V-VI	66	26/40	85.4(8.1)
Mount Sinai (N=146)	0-II	60	32/28	82(7.6)
	III-IV	42	12/30	88.8(6.6)
	V-VI	44	12/32	88.0(7.5)
Arizona (N=302)	0-II	61	40/21	80.3(8.2)
	III-IV	97	50/47	86.9(6.9)
	V-VI	144	63/81	82.3(8.5)
ROS/MAP (N=739)	0-II	151	75/76	83.6(7.2)
	III-IV	423	148/275	88.8(6.3)
	V-VI	165	46/119	89.8(5.2)

2. In the targeted bisulfite sequencing aspect of the methods it states “Due to the limited sample size of the study, we call a CpG cluster differential DNAm if the lowest p-value

from the Differential DNAm test is less than an unadjusted p-value 0.05 among all CpGs in the CpG cluster.” I was unclear what was meant by this, do they mean that provided any CpG in the cluster was nominally significant that then the cluster was deemed to be significant?

The reviewer is correct. Due to the limited sample size, to achieve meaningful comparison, we call a locus significant if at least one of the CpGs at the locus has Differential DNAm p-value less than 0.05. We have rephrased the description to make it easier-to-understand. Please see page 8:

“Due to the limited sample size, we call a CpG cluster AD-associated if at least one of the CpGs at the locus achieves unadjusted p-value for differential DNAm less than 0.05 for any of the six traits.”

- Figure 1 is low resolution and difficult to interpret, particularly for a non-expert. This figure should be improved explaining in more detail the steps taken and the outcomes.

We have completely overhauled Figure 1 in the revised manuscript, with the help from a professional graphical designer. We think Figure 1 (shown below) has been greatly improved.

4. Figure 2A is a table so should be shown as a separate table, rather than a component of figure 2.

We thank the review for the suggestion. We now changed Figure 2A to Table 1.

5. Abbreviations in supplementary table 2A are not given. A more general reader will not know what the models are by the abbreviations. Abbreviations are also not shown in other supplementary tables too and legends could be extended for clarification.

We thank the reviewer for the suggestion. We have revised accordingly. Abbreviations for machine learning methods are provided in Supplementary tables 1-6. Legends are extended to be self-explanatory.

6. Page 6 line 152 – the p value for exons is not properly given.

“ $p = 10^{-323}$ for exons respectively” was changed to “ $p < 1.00 \times 10^{-99}$ for exons respectively”. Different software packages used slightly different ways to report p-values. To be consistent, we now keep 2 significant figures for all p-values. For extremely small p-values less than 10^{-99} , we replaced their p-values by “ $p < 1.00 \times 10^{-99}$ ”. We checked other parts of the manuscript and modified accordingly .

7. In the results the authors state that the best performance among the six traits for EWASplus was for neurofibrillary tangles. Why do they think this is?

Results from the section, "*EWASplus performance on multiple cohorts*", provides the most important clues for this observation. In that section, we show that the performance of EWASplus is dependent on the power of the training data, which is a function of the effect and sample sizes, and reflected in the magnitude of the EWAS association p-values. Thus, one explanation for the best performance with neurofibrillary tangles is that it has the most strongly associated CpGs with highly significant p-values, which enables the model to have more informative training set to use to distinguish positive sites from negative ones. The biologic reason for this is likely related to greater cellular dysfunction associated with neurofibrillary tangles in model systems compared to changes from beta-amyloid.

8. The authors highlight in the discussion that a potential limitation of EWASplus is the limited number of underlying training datasets. As highlighted in the introduction there are several EWAS in AD using the 450K array aside from the ROSMAP data. It would be good to see the authors test their approach using these other datasets, many of which are publicly available. This would be important to test in other brain regions too.

We thank the reviewer for the suggestion. We agree that more tests are needed. From literature search, we found “Meta-analysis of epigenome-wide association studies in

Alzheimer’s disease highlights novel differentially methylated loci across cortex” by Smith et al. Multiple EWASs are cataloged in this study. We have reached out to the authors and they kindly shared three EWAS datasets with us. In this revision, we added all three new EWASs to our analyses. These studies were conducted on samples collected from three cohorts: London, Mount Sinai and Arizona. Among these three cohorts, London and Mount Sinai studies use prefrontal cortex, but Arizona cohort study is done on the middle temporal gyrus, a different brain region. The results from these studies are summarized in the “EWASplus performance on multiple cohorts” subsection (page 8). We also present all the performance measures in Table 4 as follows:

Cohort	Brain Tissue	AUC	AUPR	F1	Precision	Recall
London	Prefrontal Cortex	0.697	0.272	0.325	0.248	0.471
Mount Sinai	Prefrontal Cortex	0.863	0.604	0.481	0.364	0.708
Arizona	Middle temporal gyrus	0.699	0.233	0.275	0.196	0.461

From these new results, we see that EWASplus performed well, but the results are slightly worse than the results obtained from the ROS/MAP cohort, especially for the London and Arizona cohorts and for the F1 values. We think a possible explanation is that these cohorts have smaller sample sizes than ROS/MAP (N = 113, 146, 302 vs 717), which limit the power of the original EWASs. demonstrated by the fact that there are less number of CpGs show significance in these newly added EWASs. In contrast, from which cohort the samples are collected and which brain region were studied seem to have little effect on the performance of EWASplus. In other words, we believe EWASplus is applicable to all EWASs and can perform reasonably well in general.

To explore what factors could influence EWASplus performance, we conducted a regression analysis, in which for each EWAS study, the Area under the curve (AUC) of the Receiver Operating Characteristic (ROC) Curves is taken to be the response variable, and the average negative logarithm transformed p-values ranked in the top 1,000 is chosen to be the independent variable. We found a significant linear correlation between the average negative logarithm transformed p-values of top CpGs and the EWAS AUCs. Similar correlation is also found between AUPR, F1, Precision and Recall. The results are summarized in Supplementary Figure 5 and in a subsection named “Performance of EWASplus dependent on EWAS quality” in the results section.

Supplementary Figure 5.

9. Have the authors considered using the approach with EPIC array datasets? In addition could the approach be validated using publicly available WGBS datasets with matched 450K data?

We thank the reviewer for the suggestion. These are great questions. We have done a comprehensive literature search and also reached out to multiple groups but only found one EWAS study (conducted on the Brains for Dementia Research (BDR) cohort) conducted using the EPIC array with sufficient sample size. But unfortunately, when we contacted the PI, we were told that the complete EWAS data is not ready yet for sharing. They have agreed to share the data upon publication. We will run EWASplus on that dataset then.

It has been demonstrated in the literature that EPIC array is capable of producing high quality data, and the methylation measurement has good correlation with that of the 450K array. Hence, we believe EWASplus will work well on data produced by the EPIC array.

For WGBS dataset, we are unable to locate any public WGBS data available that match the 450 data we have including the ROSMAP, London, Mount Sinai and Arizona cohorts. It is definitely an important follow up work to do to validate the results of EWASplus. We will be monitoring the literature, and test on matching WGBS data when such data become available. Meanwhile, the validation study we conducted is sequencing based, which we believe that data quality is comparable to that of the WGBS. So, we expect WGBS data in the future will also validate findings from EWASplus.

10. Given that methylation profiles are tissue specific and also can change with age/environment. Could the authors comment in the discussion about the application of the method and/or results to other tissues (such as blood) and in samples with different demographics.

We thank the reviewer for this great suggestion. The reviewer is absolutely right that methylation profiles are tissue-specific and also can change with age/environment. This implies to us that the trained EWASplus model is only valid for the specific tissue type, collected from samples with certain age/environmental profile and demographics. Since the major utility of EWASplus is to expand the coverage of a specific EWAS beyond the array, this should not be a problem. But one should exercise caution when trying to extrapolate the results to any other diseases/tissue types/age/environment/demographics. We had added discussion in the manuscript in the Discussion section on page 12:

“Although the EWASplus methodology is general and can be applied to any tissue type, the methylation profiles are tissue-specific, may change with age/environment and demographics. This implies that the trained EWASplus model is only valid for the specific tissue type collected from samples with certain age/environmental profile and demographics. One should exercise caution when trying to extrapolate the results to other tissue types such as blood, or subjects with different age or environmental and demographic profiles. Since the major utility of EWASplus is to expand the coverage of EWAS beyond the array within a specific experimental dataset, this limitation will not hamper the utility of EWASplus.”

Reviewer #2 (Remarks to the Author)

DNA methylation is one of the most widely studied epigenetic changes. However, the Illumina 450K array only covers 2-3% of all CpG sites in the human genome and has known technical limitations. To fill this gap, Huang et al. proposed a novel machine learning method called EWASplus to predict the associated CpG sites at the whole genome level. Briefly, they use the soft-voting method to combine the results from four base learners (including penalized regression, support vector classifier, random forest, and XGboost). They applied EWASplus to six Alzheimer's disease (AD) related traits and validated parts of their findings experimentally. I have several comments/questions and hope the authors can consider addressing them.

1. There are several methods have been proposed to predict genome-wide DNA methylation. For example, Zhang, W. et al. *Genome Biology* 16, 14 (2015) proposed using the random forest to predict missing DNA methylation levels for all CpG sites. This seems to be highly related to the EWASplus, although they achieve the goal in a different way. The method proposed here (EWASplus) tries to divide the CpG sites into positive (i.e., signal) and negative CpG sites first and then predict the status by an ensemble method, while the method proposed in Zhang et al. tries to predict DNA methylation levels only. The author may consider comparing their method with these existing methods.

We thank the reviewer for the suggestion. We are aware of the work of Zhang et al. and have cited their paper. Following the reviewer's advice, we compared the performance of EWASplus with Zhang et al. we found EWASplus outperforms Zhang et al. method for identifying differential DNAm CpGs between AD and control samples. We think this is because predicting methylation level at individual CpG sites for individual samples is rather challenging because methylation level is dynamic, and sensitive to many factors that are difficult to control such as age, cell type proportion and experimental procedure. EWASplus bypassed this step and facilitate a direct approach to predict novel differential DNAm CpG sites. We added a paragraph in the Discussion section which is on page 10-11 and copied as follows:

“A key idea of EWASplus is that it bypasses inferring the individual-level DNAm level directly. A similar approach has been used to predict additional trait-associated genetic variants using GWAS and machine learning 35. Since our goal is to identify disease-associated DNAm CpGs rather than methylation status directly our approach avoids much complexity associated with accounting for the many factors that can influence DNAm CpG status (e.g., age, cell type proportion). This is illustrated by the performance of EWASplus compared to the modification of Zhang et al.'s method to address disease-association, which it was not originally designed to do, admittedly.”

The performance comparison figure with Zhang et al.'s method (supplementary figure 3):

Performance Comparison of EWASPlus v.s. Zhang et al. RF based method

- EWASplus directly predicts if a CpG site is associated with the trait of interest by using several features as predictors. This may induce one problem that the associations may go simply through those features. While EWAS is an association test and never imply causal, this may still count as a weakness of the EWASplus. In contrast, predicting DNA methylation levels directly and then conducting the association tests may reduce the concerns. This also backs to my Comment 2; the author may need to justify why they want to predict positive and negative CpG sites directly.

The reviewer is right about the difference of EWASplus and other methods such as the one proposed in Zhang et al. Following the reviewer's suggestion, we conducted performance comparison between EWASplus and the Zhang et al. method. Details can be found in our answer to the previous comment. Our results suggest a direct approach such as EWASplus returns better results than an indirect one (impute methylation level first for each individual sample, then conduct EWAS). The idea behind EWASplus is similar to the approach developed for GWAS-identified variants. In our previous work (Chen et al., 2016), we showed that trait-associated SNPs display a unique hallmark that may be discovered using machine learning approaches. And the unique pattern can be applied genome-wide to identify additional SNPs that associated with the trait of interest. We believe a similar idea can be applied to EWAS. We believe this is reasonable since trait-associated CpGs likely disrupt normal regulatory mechanism which are likely to located in regulatory regions in relevant tissue or cell types. These regions are recognizable by genomic and epigenomic features. From the biology standpoint, predicting if a CpG site is associated with the trait of interest using omics features as predictors is a more direct approach than a population-based approach such as EWAS. As explained in the previous comment, we think EWASplus benefits from taking such a direct approach, as evident by improved performance.

The reviewer's comment on EWASplus' prediction is well taken. The goal of EWASplus is to prioritize selected CpGs among millions of them in the genome that are not on the methylation array for further testing. For a CpG, no matter how highly ranked by EWASplus, should only be considered as "putative" in terms of trait-association unless it can be validated using experimental approaches. We have added the following sentence in the discussion section in the manuscript (page 13):

"For a CpG, no matter how highly ranked by EWASplus, should only be considered as "putative" in terms of trait-association unless it can be validated using experimental approaches."

3. The method descriptions are not very clear. For example, it is unclear how to predict the missing DNA methylation levels based on the features listed in Supplementary Table 6. The feature names in Supplementary Table 6 are not very informative. Also, do you use both cis and trans information or only use the cis information only? Another example is the soft voting method. Do you use equal weights for each base learner or taking the base learner performance into account? Without these details, it is hard to evaluate the validity of the EWASplus.

These are good questions. EWASplus only uses cis information to predict whether a locus may be differentially methylated between AD and control samples.

EWASplus uses soft-voting (assigning equal weights to the output of each learner) in the ensemble part. In general, each base learner, trained using features listed in Supplementary Table 6 (now Supplementary Data 2-10), outputs the probability of a locus to be differentially methylated for certain AD related traits, i.e., a value ranges from 0 to 1. We considered all combinations of four base learners (individual base learner; two-way cross and up to four-way cross) and average their outputs as the final prediction of the ensemble model. The best combination of the models is selected based on the predicting performance on the test dataset. This approach has an advantage over stacking and fine-tuning the ensemble weights for four base learners according to their respective performance since the model tends to overfit and lean heavily to the model with high complexity, for example, random forest and GBDT, which will further harm generalization performance.

We have revised the feature names listed in Supplementary Table 6 (now Supplementary Data 2-10) to make them easier to understand.

4. DNA methylation levels will change depends on smoking status. Many studies incorporate smoking status (or predicted smoking status) as a covariate to reduce the concern of false positives. The authors may consider report genomic control factors.

This is a good question. We agree with the reviewer that smoking status may have an impact on the DNA methylation levels. Since very few people in the ROS/MAP cohort was a smoker, the smoking status is not used in the original EWAS study. Covariates like smoking status affect the DM calling for CpGs hence the training set. But once a training set is decided, EWASplus model only utilize genome-wide features to identify new CpG sites for disease-association. Under a classification framework, the performance depends on the quality of the training data. If smoking status is considered in the array-based EWAS, there is likely less false positives in the results, so the performance of EWASplus will be better than using a training data from an EWAS in which smoking status is not used. The same applies to genomic control factors.

We agree that we would like to limit false-positive findings. The way that EWASplus is framed is as a classification problem and not a testing one; therefore, the most relevant characteristics are the precision, recall, and AUC. We think this was not made as clear as it could have been and we have added comments to those sections indicating that these parameters are traditional ways of controlling false-positive results using the ML approach we took to this issue. As the reviewer can see, we have excellent/good measures of precision, recall, AUC, which reflects a careful attention to avoid false-positive results.

5. The evaluation is based on cross-validation, which may be over-optimistic. If possible, using independent data or reserve an independent set to evaluate the performance may helpful.

This is a good point and we apologize that we did not make it clear in our original manuscript. The performance of our models reported in the current Table 1, Figure 2A, 2B are based on independent test results, or more precisely, out-of-bag estimates for test error. This is different for how we usually evaluate the validation error since the dataset under evaluation is kept untouched and the typical validation set is usually used to perform hyper-parameter tuning and hence will introduce bias in evaluating the true model performance. Here is the how we performed the evaluation:

We performed 10-fold rotations on the entire data set. For each rotation, we performed 3-fold cross-validation on the training + validation datasets (90% of the total dataset) to select the best set of hyper-parameters for each base learner. Then, we predict on the remaining one-fold (test dataset) based on four base learners (trained with best hyper-parameter just selected). The predictions are based on the mean output all possible combination of models. The test dataset is kept untouched during the hyper-parameter tuning process under the ongoing 3-fold cross-validation. We have revised the descriptions in the manuscript about the results to more accurately reflect the procedure used in this study. Please see page 17 updated section “Hyperparameter Tuning and Ensemble Model” section in the “Method” part for the detailed description.

6. Based on Figure 3A, it seems Array-based EWAS has much better performance than the proposed EWASplus. This is somewhat contradicted to Figure 2, which shows the AUC of Neurofibrillary Tangles is 0.962. Also, as EWASplus tests the whole genome’s CpG sites, using the same cutoff as before may be problematic as they test more.

The reviewer is right that performance wise, array-based EWAS has better performance than the proposed EWASplus. This is expected since differentially methylated CpGs identified by array-based EWAS are considered gold standard in our study and these CpGs are used as training data to train the machine learning model in EWASplus. And EWASplus result is aimed to be on-par with the result from array-based EWAS, so it is unlikely that prediction results from EWASplus will outperform results of the array-based EWAS.

There is no contradiction. In Figure 2, we only compare performance on CpGs on the 450K array. Since we are using array-based EWAS as training data, the AUC is 1.0 for array-based EWAS. In Figure 3A, we are testing in CpGs off the 450K array. When the training data and testing data are measured using different platform (array-based or sequencing-based), it is very reasonable to see performance drops.

Unlike regular EWAS, the EWASplus method is not a testing-based method, but rather developed under the classification framework. Therefore, the multiple testing issue faced by EWAS does not apply to EWASplus. Various measures such as AUC, AUPR, F1, precision and recall are typical performance measurements used under the classification framework which we have evaluated and presented. Normally there is no significance threshold under a classical classification framework. We simply using the top 1,000, or top 10,000 CpGs for follow-up. We have updated the text to clarify the use of the threshold. Please see page 13: “EWASplus does not provide a significance cut-off threshold since it is a supervised classification approach, not a testing-based method. In practice, one can select the threshold empirically by checking whether top CpGs identified by array-based EWAS made the cut-off.”

REVIEWERS' COMMENTS

Reviewer #1 (Remarks to the Author):

I am happy that the reviewers have addressed all of my comments.

Reviewer #2 (Remarks to the Author):

I appreciate the authors' great efforts to address my comments. I really enjoy reading your manuscript and the response letter. The quality of the manuscript has been greatly improved, and the key idea is much more clear and convincing than the original submission. I only have the following three minor questions.

1. The 2256 genomic/epigenomic profiles and functional annotation scores used in this manuscript are valuable sources and may be highly useful for the community. It would be great if the authors can share these resources such that researchers can reproduce the results in the manuscript or develop similar models for other tissues/diseases. Also, I am very curious about how to process and compile these functional annotations (such as QC steps). It would be great if the authors can provide more details (perhaps in supplementary).
2. In the response letter, you stated that "From the biology standpoint, predicting if a CpG site is associated with the trait of interest using omics features as predictors is a more direct approach than a population-based approach such as EWAS." Can you further explain this? Thank you!
3. What's the outcome for the three additional cohorts? I am curious about why the significance cutoffs are different in different datasets (It is very clear in the response letter, but I felt confused when reading the manuscript).

Response to reviews for NCOMMS-20-37924A

Below we present itemized responses to additional comments from Reviewer 2. The reviewers' comments are in regular font. Our responses are in dark blue color.

Reviewer #2

1. The 2256 genomic/epigenomic profiles and functional annotation scores used in this manuscript are valuable sources and may be highly useful for the community. It would be great if the authors can share these resources such that researchers can reproduce the results in the manuscript or develop similar models for other tissues/diseases. Also, I am very curious about how to process and compile these functional annotations (such as QC steps). It would be great if the authors can provide more details (perhaps in supplementary).

We thank the reviewer for the great suggestion. We are fully committed to share all the data produced from this study without any restriction. Per the reviewer's suggestion, we have uploaded the preprocessed data (all 2,256 features) to figshare. The link is https://figshare.com/collections/Dataset_collection_of_EWASplus/5430207. We have included in the Data Availability section of the manuscript. Please see page 20 of the revised manuscript.

A detailed guide on how to generate the annotations, including scripts, are provided at our public github repository (<https://github.com/xsun28/EWASplus/tree/remastered>). Steps 0-2 outline the steps from compiling, processing, and assigning genomic features to genomic loci.

Following the reviewer's advice, we added the description of how we process the features. Since the Supplementary Information file is restricted to figures and tables, we put the detailed description in the description page of the figshare dataset (https://figshare.com/articles/dataset/Genomic_Epigenomic_features/14614112?backTo=/collections/Dataset_collection_of_EWASplus/5430207). The description is copied below: "The main type of features we used is the read counts from various sequencing experiments mentioned above. We firstly downloaded the alignment files (bam files) from ENCODE website. For the sake of consistency, we convert the genomic coordinates to hg19 human reference genome assembly if the original mapping is done using hg38. After that, we retrieved the genomic location information of each read from the alignment files and then count the number of reads located in each pre-defined 200bp bin across the whole human genome. For the experiment with multiple technical replicates, we merged the read counts into a single feature using their mean values. For other features, such as annotation scores, which are typically precomputed at basepair level, we use the score assigned to the exact CpGsite as the feature value."

2. In the response letter, you stated that "From the biology standpoint, predicting if a CpG site is associated with the trait of interest using omics features as predictors is a more direct approach than a population-based approach such as EWAS." Can you further explain this? Thank you!

What we meant to say is that EWAS tests for trait associated CpG based on methylation difference among case and control samples, and, by itself, it does not provide a biological explanation for the association. On the other hand, EWASPlus predicts trait associated CpGs based on omics features, which provides a potential biological reason for the CpG association with the trait.

3. What's the outcome for the three additional cohorts? I am curious about why the significance cutoffs are different in different datasets (It is very clear in the response letter, but I felt confused when reading the manuscript).

We thank the reviewer for the comment. The outcome of the three additional cohorts is Braak staging (treated as continuous variable). We are sorry this is not made clear. In the revised manuscript, we added "In all three studies, Braak staging (treated as a continuous variable) is used as the trait in the EWAS studies, as described in Smith et al.²⁵" Please see page 8.

To better explain why the significance cutoffs are different in different datasets, we added a paragraph in the discussion section of the revised manuscript. Please see page 13 and copied below:

"Deciding on the number of significant EWAS CpGs to include in training is a tradeoff between the quantity and quality of the training set in EWASplus. Thus, the significance threshold for each EWAS should be decided based on the effect size and sample size of the EWAS. Future work should examine the utility of including different thresholds and use cross-validation to select the desired significance cutoff."